# Assessing the Effects of Rainfall Intensity and Hydraulic Conductivity on Riverbank Stability

**Toan Thi Duong** [1,*]**, Duc Minh Do** [1] **and Kazuya Yasuhara** [2]

[1]  Department of Geotechnics and Infrastructure Development, VNU University of Science, Vietnam National University, Hanoi 100000, Vietnam; ducgeo@gmail.com
[2]  Institute for Global Change Adaptation Scienceicas (ICAS), Ibaraki University, 2-1-1 Bunkyo, Mito, Ibaraki 310-8512, Japan; kazuya.yasuhara.0927@vc.ibaraki.ac.jp
*  Correspondence: duongtoan109@gmail.com; Tel.: +84-093-454-3261

**Abstract:** Riverbank failure often occurs in the rainy season, with effects from some main processes such as rainfall infiltration, the fluctuation of the river water level and groundwater table, and the deformation of transient seepage. This paper has the objective of clarifying the effects of soil hydraulic conductivity and rainfall intensity on riverbank stability using numerical analysis with the GeoSlope program. The initial saturation condition is first indicated as the main factor affecting riverbank stability. Analyzing high-saturation conditions, the obtained result can be used to build an understanding of the mechanics of riverbank stability and the effect of both the rainfall intensity and soil hydraulic conductivity. Firstly, the rainfall intensity is lower than the soil hydraulic conductivity; the factor of safety (FOS) reduces with changes in the groundwater table, which is a result of rainwater infiltration and unsteady state flow through the unsaturated soil. Secondly, the rainfall intensity is slightly higher than the soil hydraulic conductivity, the groundwater table rises slowly, and the FOS decreases with both changes in the wetting front and groundwater table. Thirdly, the rainfall intensity is much higher than the soil hydraulic conductivity, and the FOS decreases dominantly by the wetting front and pond loading area. Finally, in cases with no pond, the FOS reduces when the rainfall intensity is lower than hydraulic conductivity. With low hydraulic conductivity, the wetting front is on a shallow surface and descends very slowly. The decreasing of FOS is only due to transient seepage changes of the unsaturated soil properties by losing soil suction and shear strength. These obtained results not only build a clearer understanding of the filtration mechanics but also provide a helpful reference for riverbank protection.

**Keywords:** riverbank stability; rainfall intensity; hydraulic conductivity

## 1. Introduction

Rainfall is one of the main factors causing slope failure in tropical areas. The effects of rainfall properties on slopes have been studied in large amounts of research that have analyzed slope stability by both investigation and simulated models. By the simulation of different conditions of slopes and rainfall boundaries, previous research has indicated the changes of soil unsaturated properties, such as the reduction of suction, shear strength, and the increasing of hydraulic conductivity and pore-water pressure [1–20]. Finally, these changes cause slope failure. Before rain, the slope area above the groundwater table is considered as being in a partly unsaturated and dry state near the surface [1–4]. The unsaturated area reduces by rainfall water infiltration during and after the rainfall event. The change of the area from unsaturated to saturated is caused by the advancement of the wetting front from the surface [4,5,9,13,21–23] and groundwater table from depth [6–8,10–20]. Those processes were found to be the primary factors controlling the instability of slopes due to

rainfall and were greatly affected by rainfall intensity (RI) and soil properties, especially by unsaturated soil hydraulic conductivity [4,17–36]. The unsaturated soil hydraulic conductivity (HC) controls the transient seepage, the depth of rainfall infiltration, the changes in pore pressure during the rainfall event, and finally, affects the FOS. The effects of soil hydraulic conductivity on slope stability in the rainy season are usually assessed from the point of view of three topics: (1) the effects of changes in the value of soil hydraulic conductivity and rainfall intensity [17–30]; (2) the soil anisotropic HC and hydraulic hysteresis [31–35]; (3) the failure delay phenomenon due to the defense of HC and RI [36]. This research focuses on the overview and analysis of the first of these topics.

The effects of the HC on the FOS were specifically simulated by [25,26]. Those research works reported that high hydraulic conductivity led to rapid saturation. Additionally, infiltration causes the wetting front to quickly shift downward. This shift causes water to contact the underlying impermeable soil, leading to a rapid rise in pore-water pressure and the formation of a perched water table. The slope reaches full saturation and experiences a reduction in soil resistance. Consequently, the factor of safety rapidly decreases [26]. High-intensity rainfall has large effects on the slope if the soil slope has high hydraulic conductivity (Ks $\geq 10^{-4}$ m/s) and when the slope has poorly drained soils (i.e., Ks $\leq 10^{-6}$ m/s) [25].

The simulated analyses were also carried out with different boundaries of rainfall. When building the relationship of rainfall intensity and unsaturated hydraulic conductivity by the ratio of RI/HC, the previous research analyzed the RI = HC [5], RI $\geq$ HC [19,22]; and RI < HC [8,17,21,23,26,29] cases. Setting up a boundary with RI = HC or RI $\geq$ HC and no pond [5,19,22], the development of a wetting front from the crest of the slope and the reduction of soil suction during rainfall were found to dominantly affect the slope stability. The development of the defense of the wetting front under increased rainfall intensity shortens the time required for the wetting front to reach the pore-water pressure and moisture content sensors. The growing of groundwater has not been mentioned much. The same trend was also found in [17,21,23], which had the boundary of RI < HC. The larger the coefficient of permeability is, the greater the depth of the wetting front is. The opposite mechanism is seen in [8,29] with a boundary of small RI and high initial saturation and hydraulic conductivity, and the pore-water pressure increases gradually from the deep part to the crest of the slope [8,29]. For a slope with a larger HC, the slope failures possibly take place under rainfall with a shorter duration and a greater intensity [8]. Those research works also concluded that when the rainfall intensity is greater than the saturated hydraulic conductivity of the surface soils, a runoff occurs along the slope surface [8,22].

Although those previous research works had unobvious objectives focusing on building the effects of hydraulic conductivity, the results clearly showed that the slope stability was significantly affected by the soil hydraulic conductivity and the rainfall intensity boundary. The ratios of RI/HC were divided into three cases; however, each previous research work only concentrated on one of those cases. Moreover, there was a difference regarding the development of the pore-water pressure by the rising groundwater table [8,29] or by a wetting front from the surface [21,23] when those analyses had the same initial conditions. Further research should cover all three cases of different RI/HC ratios.

A riverbank is a special slope that relates closely to hydraulic dynamics, not only from rainfall infiltration from the surface concerning the space above the slope, but also significantly from the fluctuation of the river water level and groundwater table [27,37–43]. Past research works assessed the effects of rainfall on riverbank stability using the indirect process of transient seepage due to water level changes and assumed that rainfall has no effect on the riverbank surface.

By reviewing previous papers, it was obvious that the hydraulic conductivity has great effects on slope stability during rainfall events. However, some limitations were found in the reviewed research. The effects of the different RI/HC values were not specific to a researcher, and there was a difference in the previous discussion on seepage mechanics, such as the changes in the wetting front and the groundwater table. Moreover, research on the effects of hydraulic conductivity on riverbank slope stability was performed in only a few papers. Therefore, the objective of this paper is to build

the mechanics of riverbank failure with different rainfall infiltration and soil hydraulic conductivity models. The numerical analyses by the GeoSlope program with both SEEP/W and SLOPE/W moduli are applied in this study.

## 2. Materials and Methods

The case study is a riverbank area in the Red River in Hanoi city, Vietnam. The Red River originates in Yun-nan, China, and flows through northwest Vietnam. The total length of the Red River in Hanoi is about 90 km with about 40 km of riverbank inside the urban area of Hanoi, where the population density and density of houses built near the natural riverbank are highest. These areas are located in high river terraces, outside the river dike, and near natural banks. In the annual rainy season, the destruction of local houses, roads, and land-use along the Red River bank has occurred.

The selected riverbanks are almost natural banks, and some are supported in the toe by vegetation. The riverbanks in these locations have either collapsed or have high failure potential due to fluctuations in the river water level, river water flow stress, rainfall, and human activities. Figure 1 shows the road near the riverbank (left), and the natural riverbank (right), which was broken in the rainy season of 2016 and 2017.

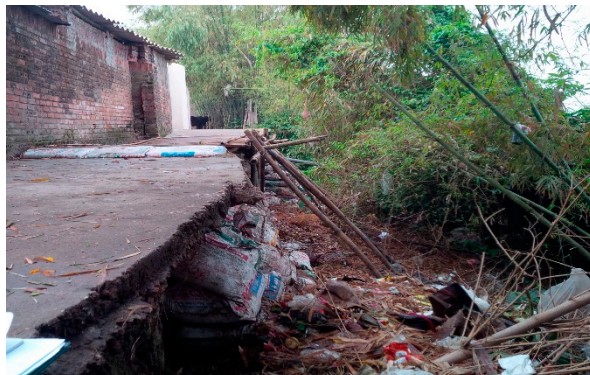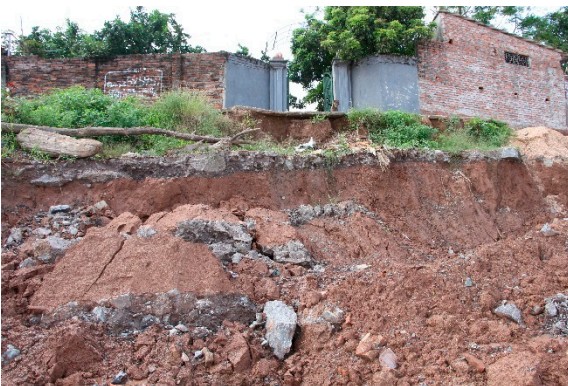

**Figure 1.** Some current problems near the riverbank in the study area.

### 2.1. Field Investigations and Soil Properties

The field investigation was performed during both the dry and the rainy seasons to describe the status of the riverbank and the river water level changes. Field data measurement and collection included bank geometry (i.e., height, slope), alluvial area, current river water level, and soil samples. The monitoring data, which included groundwater level, river water level fluctuation, and rainfall, were also collected from the National Meteorology Station.

The soil properties included soil physical properties such as water content, density, and grain size. The saturated hydraulic conductivity and shear strength were obtained in the geotechnical laboratory of VNU University of Science, Vietnam National University, Hanoi. The unsaturated soil properties, such as soil suction, were determined by using a pressure plate apparatus in the geotechnical laboratory of Ibaraki University, Japan, as shown in [42]. Based on the soil samples collected along the riverbank in the Hanoi area and the soil properties experiment, the riverbank in this area is quite homogeneous, with a silt or silty-clay layer in the bank layer and fine sand from the toe of the riverbank to the sediments. The silt layer is composed of less than 20% fine sand, 30–70% silt, and 10–30% clay. This paper selects one section of the soil bank to analyze riverbank stability, as shown in Table 1. Figures 2 and 3 present the suction and unsaturated hydraulic conductivity, respectively. The unsaturated shear strength and hydraulic conductivity were estimated by using the models of Vanapalli (1996) [44] and Van Genuchten (1980) [45] in the GeoSlope program [46,47].

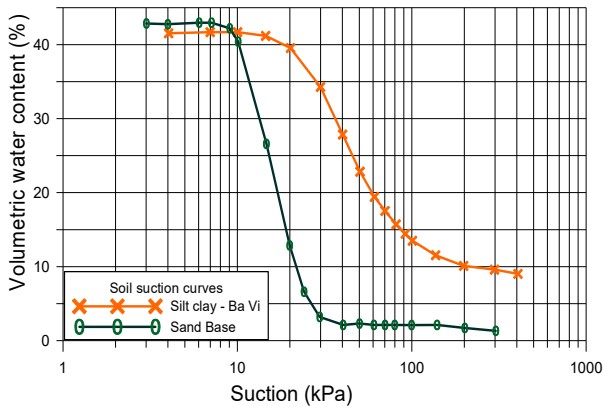

**Figure 2.** Soil–water characteristic curves.

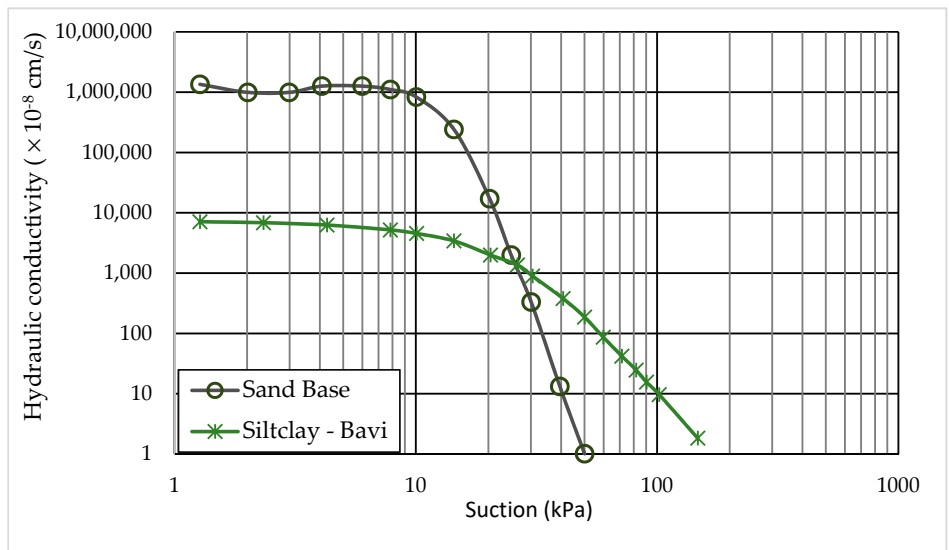

**Figure 3.** The unsaturated hydraulic conductivity curves.

**Table 1.** Soil properties used in riverbank stability analysis in Ba Vi area.

| Riverbank Soil Properties | | Silt-Riverbank Soil | Fine Sand Base |
|---|---|---|---|
| | Depth of layer | 10–1.5m | From 1.5 m to depth |
| | Coarse sand: 1–0.25 mm | | 18.38 |
| Grain size (%) | Fine sand: 0.25–0.075 mm | 13.46 | 80.91 |
| | Silt: 0.075–0.005 mm | 71.03 | 0.71 |
| | Clay < 0.005 mm | 15.51 | 0 |
| Natural water content (%) | | 18.5 | 4.72 |
| Dry density (kN/m$^3$) | | 15.0 | |
| Specific gravity | | 2.62 | 2.68 |
| Liquid limit (%) | | 34.5 | |
| Plastic limit (%) | | 21 | |
| Liquid index | | 1.23 | |
| Soil classification | | ML | SP |
| Saturated volume water content | | 42 | 29 |
| Air-entry value (kPa) | | 20.05 | 9.3 |
| Residual suction (kPa) | | 90 | 25 |
| Residual volumetric water content (%) | | 10 | 4 |

**Table 1.** *Cont.*

| Riverbank Soil Properties | Silt-Riverbank Soil | Fine Sand Base |
|---|---|---|
| a | 28.39 | 10.33 |
| n | 4.205 | 18.89 |
| m | 0.72 | 0.53 |
| Max slope | 1.33 | 2.58 |
| Hydraulic conductivity (cm/s) | $7.39 \times 10^{-5}$ | $1.69 \times 10^{-2}$ |
| Cohesion force ((kPa) | 5.0 | 0 |
| Internal friction angle (o) | 32 | 30 |

*2.2. Numerical Model Framework*

This paper uses the commercial GeoSlope program (GeoSlope International Ltd.) [46–49] as a numerical model to analyze riverbank stability. GeoSlope is one of the most useful and widely used programs in slope and riverbank stability analysis in many kinds of research [1–5,12,14,15,24–26]. The present paper uses a pair of analyses of transient seepage in SEEP/W and slope stability in SLOPE/W in the GeoSlope program.

In general, by using the SEEP/W, the mechanism describes specifically the transient seepage by rainfall infiltration. The obtained results from SEEP/W, which include pore-water pressure distribution and changes in soil properties, become the input data for the next modulus SLOPE/W for analyzing the riverbank stability. The result of the FOS indicates the riverbank stability when FOS is higher than one. The results and discussion focus on building the relationships between the FOS and the different conditions of initial saturation, rainfall intensity, and soil hydraulic conductivity.

The field investigation, laboratory testing, and monitoring of the support input data included three factor groups: the riverbank geometry, the soil properties, and the hydraulic conditions (such as the river water level and rainfall intensity). The riverbank stability analysis was performed for different initial saturation conditions, soil hydraulic conductivity, and rainfall intensity. The initial saturation conditions were set up by the initial river water level and the capillary height or the maximum negative head. In SEEP/W, the maximum negative head can be used to build the assumption of the predetermined negative pore pressure profile, and then the saturation condition can be set up. Other simulated factors such as the soil hydraulic conductivity and rainfall intensities were installed as the difference between the soil properties and boundaries. The conditions of the riverbank, hydraulics, and boundaries for different scenarios of the initial saturation and rainfall intensity are described in more detail below.

2.2.1. Riverbank Geometry and Hydraulic Boundary Condition

This paper uses a riverbank configuration with the initial conditions shown in Figure 4. The riverbank has a slope angle of 52 degrees and a height of 10 m. The riverbank is homogeneous silty soil with a sand layer underneath. The analyzed soil properties are shown in Table 1 and Figures 2 and 3.

The boundary conditions of river water level (RWL) and rainfall intensity were set up in the SEEP/W model. Using the riverbank configuration shown in Figure 4, the initial RWL was the boundary in the river site from the bottom to 3 m, and the function RWL–time was the boundary of the entire lateral riverbank. The rainfall was the boundary on the surface of the riverbank. Both scenarios with and without the function "potential seepage face review" were used with the rainfall boundary. Without the function "potential seepage face review", the rainwater would infiltrate into the soil as long as the rainwater exceeded the infiltration capacity; under these conditions, a pond would present above the riverbank surface when there was excess rainwater. This condition often occurs at some riverbank sites that have a pond or low areas near the riverbank and during floods. If the "potential seepage face review" was included, the pond would not exist, and the excess rainwater would run off.

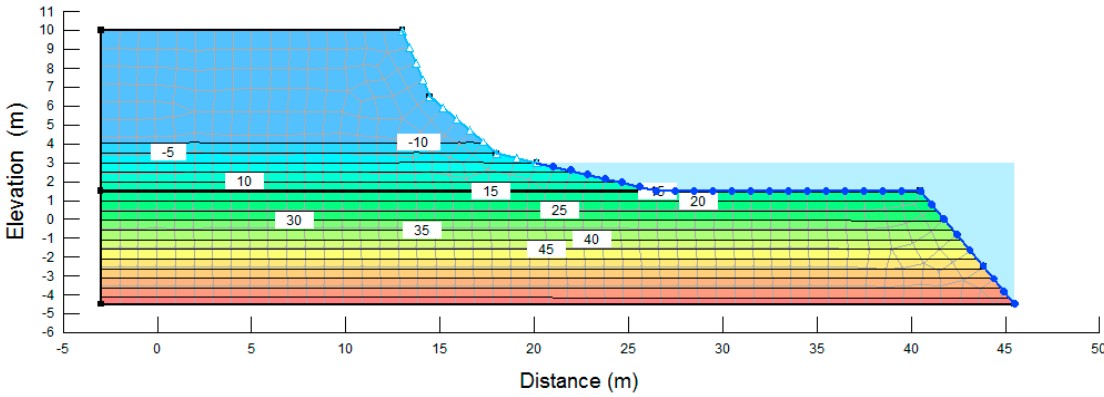

**Figure 4.** The initial riverbank configuration.

The rainfall intensity was determined based on daily monitoring data. The rainy season is from mid-June to mid-September, and high rainfall often occurs in August. Figure 5 shows the variation in the rainfall intensities (RI, mm/h) of several rainy days in the rainy month (August 2016), and Figure 6 shows the changes in daily rainfall and RWL monitored during the rainy season from 1 July to 31 August where some sites in the Red River bank broke. It can be seen that the rainfall intensity ranged from 0–50 mm/h. To simulate the effect of rainfall intensity on riverbank stability, three rainfall intensities were used: RI = 10 mm/h; RI = 30 mm/h; and RI = 50 mm/h. Based on the RWL data, the initial RWL was set as 3 m.

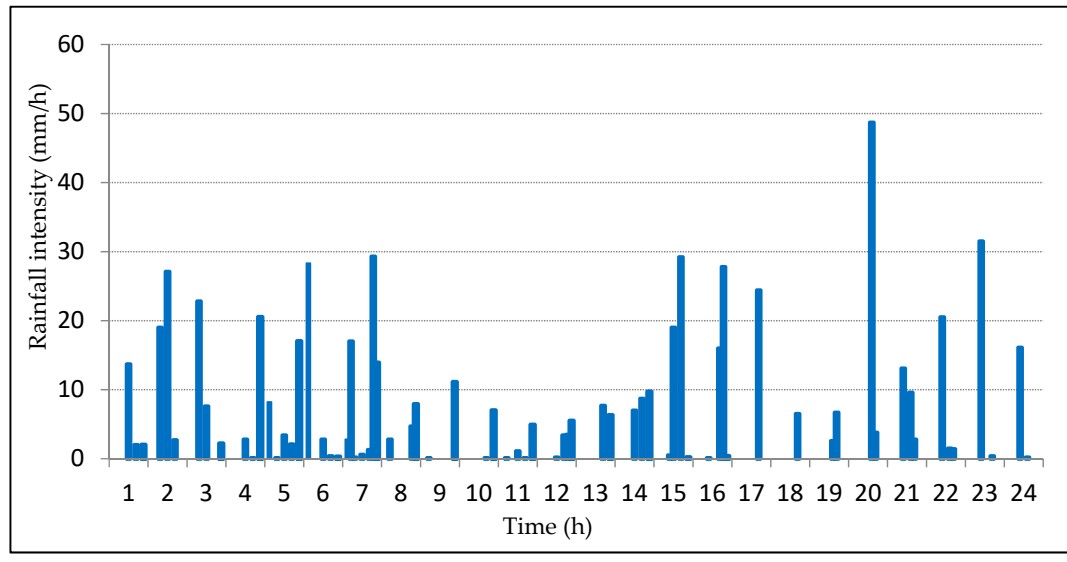

**Figure 5.** The hourly rainfall with high rainfall intensity for some days in August 2016.

The initial pore-water pressure and saturation conditions were set up by using the function of the maximum negative pressure head in SEEP/W. This function was built with knowledge of the pore pressure in unsaturated soil. Soil regions above the groundwater table were divided into two sub-regions: a dry zone near the surface and a partly saturated zone near the groundwater table. The pore-water pressure graph was linear and negatively sloped from the groundwater table to the maximum negative head. This meant that the negative pore-water pressure near the surface may have become too high in the dry zone. In fact, the soil was never completely dry and always retained some amount of water. The small surface flux had the effect of changing the pore-water pressure profile. Figure 7 shows a pore-water pressure profile with a non-dry surface condition in which the negative pore-water pressure was negatively linearly sloped to a maximum negative pore-water pressure and remained constant at a value in response to soil water content. The magnitude of the maximum

negative pore-water pressure was dependent on the shape of the hydraulic conductivity function, and to a lesser extent on the rate of infiltration. In SEEP/W, setting the maximum negative head can indicate the field pore pressure profile. Based on the investigation and experimental data of the soil water content and soil suction curve, the value pore pressure or saturation degree could be determined.

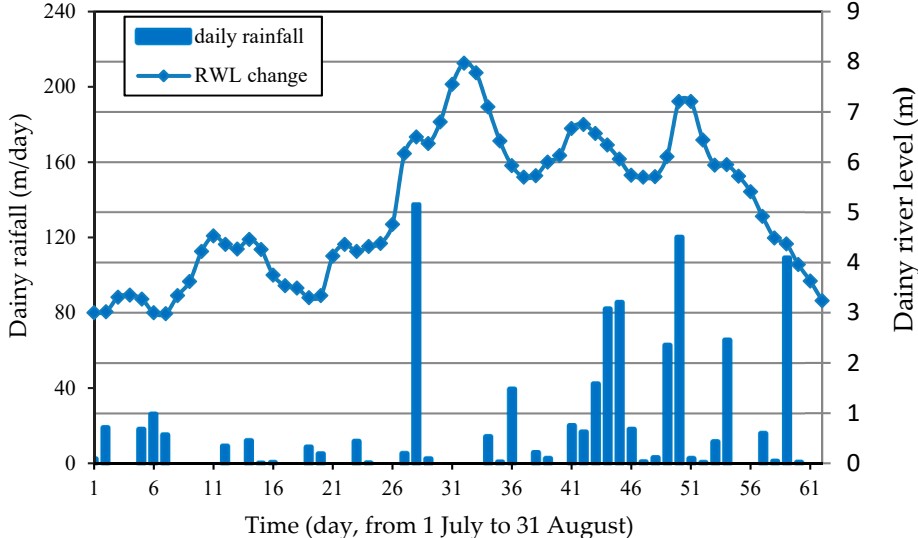

**Figure 6.** Daily river water level (RWL) and daily rainfall from 1 July to 31 August 2016.

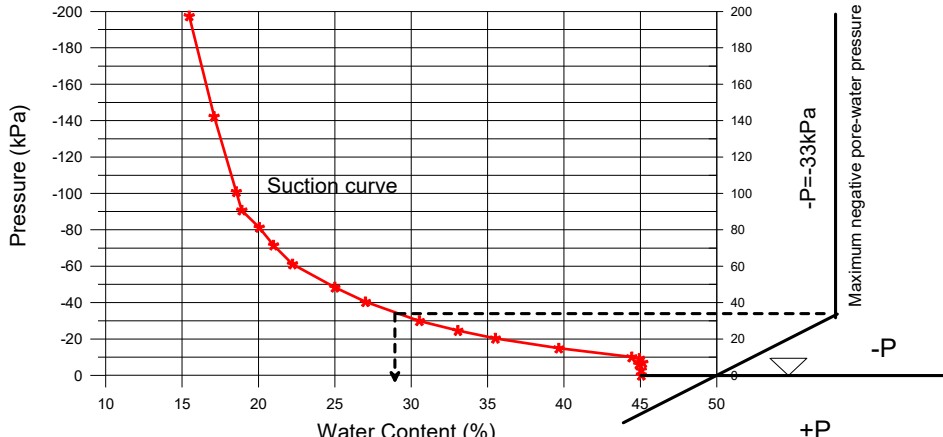

**Figure 7.** Calculation pore-pressure condition by saturation condition.

To simulate the effect of the initial pore pressure and saturation conditions, we set the different initial negative pore pressures to 15 kPa and 33 kPa, respectively. At those pressures, the relative soil water contents were 41% and 33%, and the saturation degrees were 87% and 70% respectively. Those were average experimental values in the beginning of the rainy season and in the rainy season.

The initial riverbank soil had a saturation hydraulic conductivity of Ks = 7.39 × $10^{-5}$ cm/s (Table 1). To simulate the effects of soil hydraulic conductivity on the processes of rainfall infiltration and riverbank stability, three values of saturation hydraulic conductivity were used to represent the initial saturated hydraulic conductivity: Ks = 7.39 × $10^{-3}$ cm/s, Ks = 7.39 × $10^{-4}$ cm/s, and Ks = 7.39 × $10^{-5}$ cm/s. A hydraulic conductivity of Ks = 7.39 × $10^{-3}$ cm/s was considered to show high conductivity (cases $H_i$); Ks = 7.39 × $10^{-4}$ cm/s was considered medium hydraulic conductivity (cases $M_i$); and Ks = 7.39 × $10^{-5}$ cm/s was considered low hydraulic conductivity (cases $L_i$). Figure 8 shows three unsaturated hydraulic conductivity curves that correspond to the three

hydraulic conductivity values listed above and to the same suction properties for silty soil in the Red River bank.

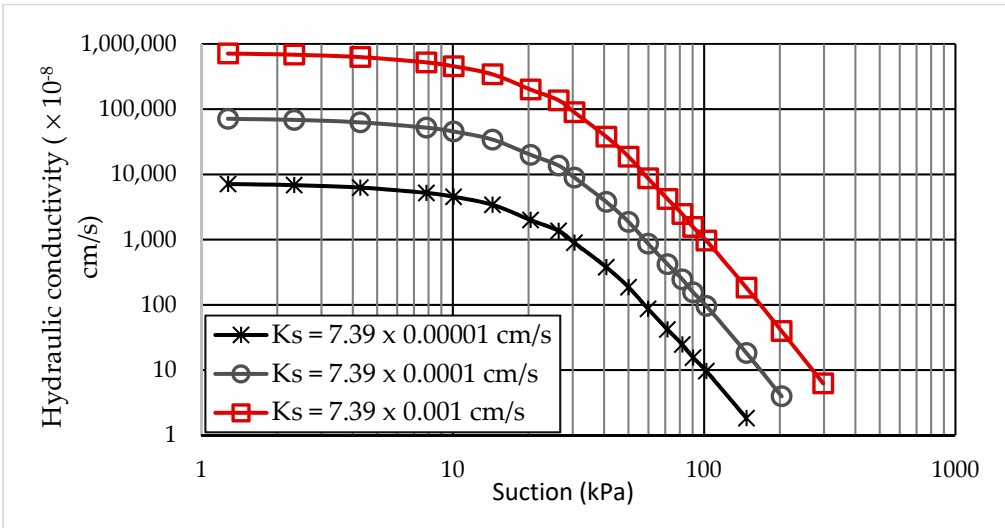

**Figure 8.** The different unsaturated hydraulic curves used in analyses.

### 2.2.2. Cases Used in Analysis

Table 2 describes the numbers of cases we analyzed to simulate the riverbank stability in the different initial suction conditions or negative pressure, the rainfall intensity, and the hydraulic conductivity, in which cases H-1-1, H-1-2, H-1-3 were the names of case studies with high saturation hydraulic conductivity (Ks = 7.39 × $10^{-3}$ cm/s) at the initial saturation degree of 70% at three rainfall intensities of 10 mm/h, 30 mm/h, and 50 mm/h, respectively. Using this labeling for cases, there were 18 analyzed cases, as shown in Table 2.

**Table 2.** Cases used to analyze Ba Vi riverbank.

| Saturated Degree (%) | Vol. Water Content (%) | Soil Suction (kPa) in Ba Vi | The Unsaturated Hydraulic Conductivity (HC, cm/s) Ba Vi | Name of Cases Versus HC | Name of Cases | | | The Ratio RI/HC at | | |
|---|---|---|---|---|---|---|---|---|---|---|
| | | | | | 10 mm/h | 30 mm/h | 50 mm/h | 10 mm/h | 30 mm/h | 50 mm/h |
| 70 | 33 | 33 | $7 \times 10^{-4}$ | H-1 | H-1-1 | H-1-2 | H-1-3 | 0.38 | 1.19 | 1.85 |
| 87 | 41 | 15 | $3 \times 10^{-3}$ | H-2 | H-2-1 | H-2-2 | H-2-3 | 0.09 | 0.28 | 0.46 |
| 70 | 33 | 33 | $7 \times 10^{-5}$ | M-1 | M-1-1 | M-1-2 | M-1-3 | 3.85 | 11.86 | 18.57 |
| 87 | 41 | 15 | $3 \times 10^{-4}$ | M-2 | M-2-1 | M-2-2 | M-2-3 | 0.93 | 2.78 | 4.63 |
| 70 | 33 | 33 | $7 \times 10^{-6}$ | L-1 | L-1-1 | L-1-2 | L-1-3 | | | |
| 87 | 41 | 15 | $3 \times 10^{-5}$ | L-2 | L-2-1 | L-2-2 | L-2-3 | 9.26 | 27.78 | 46.30 |

Where rainfall intensities (RI) = 10 mm/h = 2.7 × $10^{-4}$ cm/s; RI = 30 mm/h = 8.3 × $10^{-4}$ cm/s; RI = 50 mm/h = 1.3 × $10^{-3}$ cm/s.

## 3. Results

### 3.1. Effects of the Initial Saturation Condition

Figure 9 shows the results of FOS versus time with the saturation degrees of 70% and 87% and RI = 10 mm/h. These were cases H-1-1, H-2-1, M-1-1, and M-2-1. In these cases, the rainfall intensity was lower than the hydraulic conductivity when the saturation degree was 87%, and the rainfall intensity was slightly higher than the hydraulic conductivity when the saturation degree was 70%. Most of the rainfall water infiltrated into the soil and caused the increase of pore-water pressure by the raising of the groundwater table. The FOS results indicated the effect of the saturation condition and hydraulic conduction on riverbank stability. The higher saturation condition and higher hydraulic conductivity caused higher pore pressure; then, FOS decreased more quickly and obtained a lower value (Figure 9).

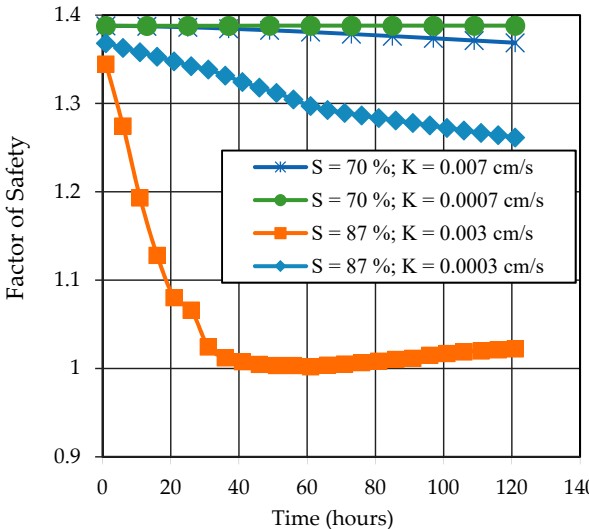

**Figure 9.** The factor of safety (FOS) results at an RI = 10 mm/h and with a different saturation degree and hydraulic conductivity.

When S = 87%, the FOS had obviously different trends that depended on the rainfall intensity and soil hydraulic conductivity. However, it was an insignificant change in the case of the analysis where S = 70%. Therefore, the below results and discussion mention only the case where S = 87 %.

### 3.2. Effects of Rainfall Intensity and Hydraulic Conductivity on Riverbank Stability

Firstly, both the rainfall intensity and rainfall accumulation affected the FOS. With higher rainfall intensity and rainfall accumulation, the FOS decreased to a lower value (Figure 10). That result was also found in most of the previous research that mentions rainfall intensity [21–30]. Figure 10A1,A2,B1,B2,C1,C2 shows the changes of FOS not only by rainfall intensity and rainfall accumulation but also by soil hydraulic conductivity in cases with a pond and no pond on the surface. In Figure 10, the red dashed line where FOS was equal to one on the graphs meant that riverbank failure occurred. The effects of hydraulic conductivity on the FOS were as follows:

When Ks = $7.39 \times 10^{-3}$ cm/s, the change in FOS had the same trend in both boundary cases. In three cases of different RIs, the FOS decreased at the same rate from the beginning of the rainfall event to approximately 30 h. After 30 h of rain, the FOS varied with the different rainfall intensities. The higher the RI was, the lower the FOS was. The riverbank was stable when RI = 10 mm/h; however, riverbank failure occurred after 36 h with RI = 50 mm/h, and after 40 h with RI = 30 mm/h. The results had the same trend as those obtained by Rahimi et al [25]. There, the FOS also decreased rapidly in the beginning of the rain event, and then FOS insignificantly changed after a threshold of RI. Once the RI reached the threshold RI, the FOS did not change any more due to an excess of rainwater run off [25].

For Ks = $7.39 \times 10^{-4}$ cm/s, the range of FOS in cases with a pond was from 1.36 to 0.39 (Figure 10B1), and from 1.39 to 1.03 in cases with no pond (Figure 10B2). In both cases, the FOS change was insignificant with an RI of 10 mm/h; however, the FOS decreased rapidly after 55 h and 110 h where RI = 50 mm/h and RI = 30 mm/h, respectively. In the case with a pond, the riverbank failure occurred at 60 h and 115 h with RI = 50 mm/h and RI = 30 mm/h, respectively. In the case with no pond, riverbank failure only occurred when RI = 50 mm/h.

For Ks = $7.39 \times 10^{-5}$ cm/s, the FOS obviously decreased in the rainfall boundary with a pond (Figure 10C1), but that changed only slightly in the case of a boundary with no pond (Figure 10C2). In the case with a pond, the FOS decreased as the rainfall intensity and accumulation increased from 1.38 to 0.2. At higher rainfall intensity, riverbank failure occurred quickly. Riverbank failure occurred after 16 h and 26 h for rainfall intensities of 50 mm/h and 30 mm/h, respectively. In the case of no

pond, the FOS decreased indistinguishably in small ranges with all rainfall intensities from 1.37 to 1.35, and the riverbank was stable after five days of rain.

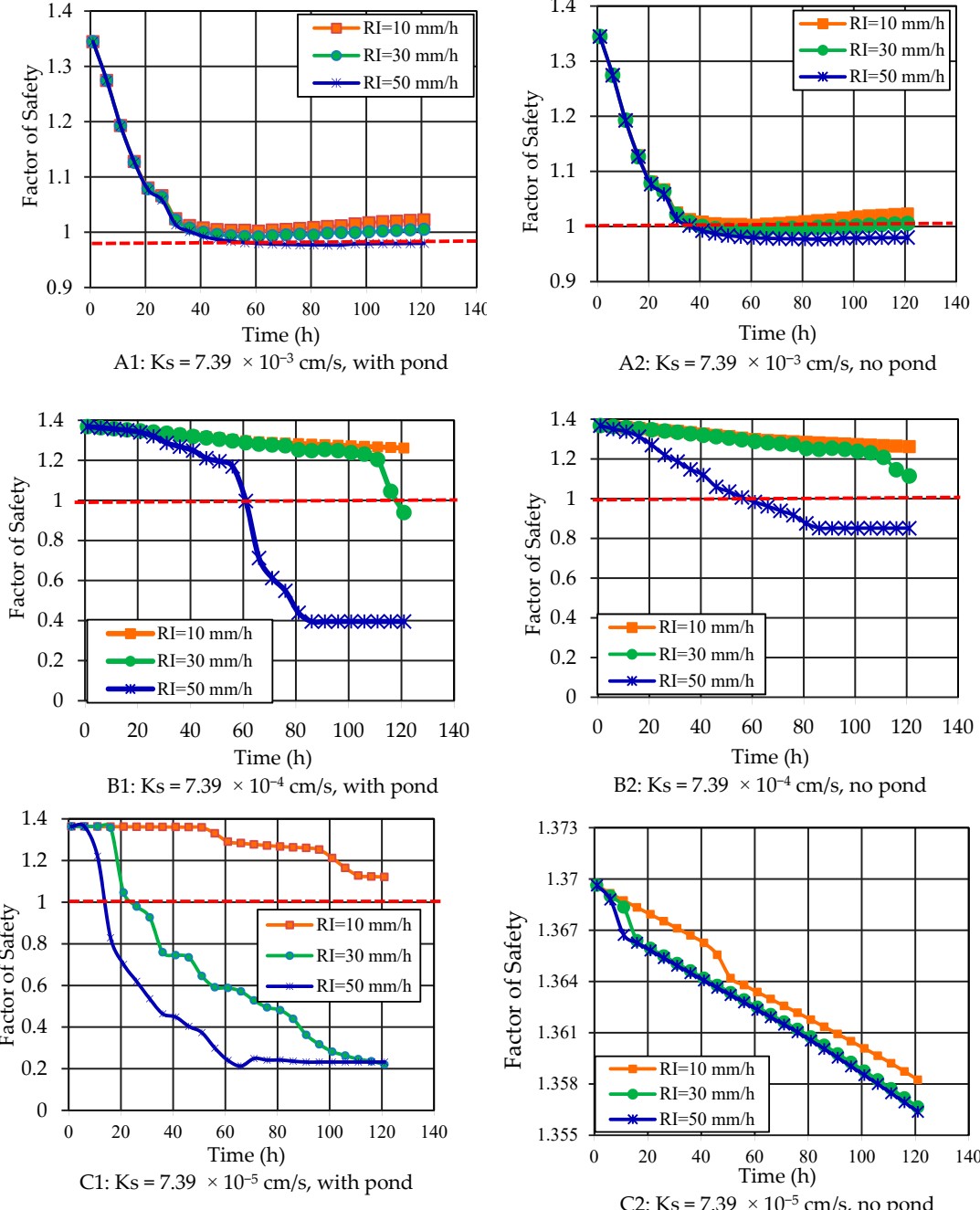

**Figure 10.** The change of FOS with different rainfall intensities and hydraulic conductivity with a pond and with no pond.

These results indicated that the drainage condition (pond or no pond) greatly affected the change of FOS. In the case of no pond, the present results regarding the change of FOS with different RIs and HC were the same as the results obtained in the same condition with no pond in [21–23,27–30]. The potential of riverbank failure was higher when the riverbank had higher hydraulic conductivity. On the contrary, the potential of riverbank failure with low hydraulic conductivity was higher than that with high hydraulic conductivity when a pond was present on the surface. With a pond on the surface, the excess rainwater not only built a loading pressure but also created a wetting front;

then, the FOS decreased more quickly. With HC < RI, the FOS had a lower value in soils with lower hydraulic conductivity (Ks = $3 \times 10^{-5}$ cm/s) than in soils with higher hydraulic conductivity (Ks = $3 \times 10^{-4}$ cm/s) because the excess rainwater in the first case was much more than in the latter case. In general, it is rare to have a pond on the surface of a riverbank or slope. However, when the riverbank slope has a high saturation degree and the soil suction is completely lost, any charge-loading in the surface will cause an unbalanced loading and riverbank slope failure to occur.

*3.3. Effects of Rainfall Intensity and Hydraulic Conductivity on the Water Infiltration Mechanism*

The mechanics of rainfall infiltration for different cases of RI and HC are indicated here by the changes of pore-water pressure and the wetting line. Pore-water pressure increased with the raising of the groundwater table, and the wetting front was linearly propagated from the surface and separated the saturated area and the initial unsaturated area by rainwater infiltration.

In the case with a pond on the slope surface, the rainfall infiltration occurred in both processes: (1) the pore-water pressure or the groundwater table rose, and (2) the wetting front propagated. The changes in the groundwater table or the wetting front depended on the rainfall intensity and the soil hydraulic conductivity.

The rainfall infiltration caused the raising of the groundwater table when the hydraulic conductivity was higher than the rainfall intensity. Figure 11 shows the groundwater table versus raining time for the case H-1-1, which had an RI/HC = 0.09. The rainwater infiltrated and moved quickly into the soil and could raise the groundwater table. The transient seepage area above the groundwater may not even have obtained full saturation, as shown in [17]. The cases of H-1-1, H-2-2, H-2-3, and M-1-1 with the ratios of RI/HC = 0.38; 0.28, 0.46, and 0.93, respectively, had the same mechanics. In these cases, an increase of pore-water pressure was the main reason causing the decrease of the FOS and riverbank failure. These results had the same trend as shown in [8,29] when the soil hydraulic conductivity was higher than the rainfall intensity.

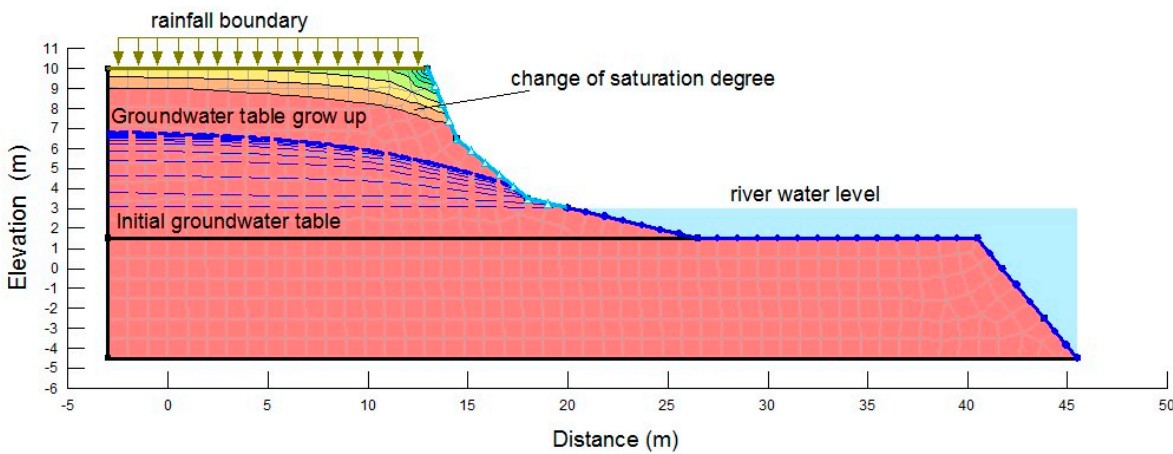

**Figure 11.** The raising of the groundwater table when the RI/HC is less than one.

The second process, in which the rainfall infiltration caused the wetting front to descend, occurred when the hydraulic conductivity was lower than the rainfall intensity or the ratio of RI/HC was higher than one (i.e., the cases of M-2-2, M-2-3, L-2-1, L-2-2, and L-2-3). With a high ratio of RI/HC or low hydraulic conductivity, as in cases L-2-1, L-2-2, and L-2-3, the rainwater infiltrated very slowly into the riverbank soil and did not cause a change in groundwater. Because of the slow transient seepage and the high rainfall intensity, an amount of excess water created a pond on the surface. When a pond was present, the wetting line appeared and descended deeper into the riverbank soil as rainfall accumulation increased. Figure 12 shows the wetting front descending in the case of low hydraulic conductivity, i.e., Ks = $7.39 \times 10^{-5}$ cm/s, with the pond above the riverbank surface when the rainfall flux exceeded the soil infiltration capacity. The increased wetting depth caused the fully saturated

zone to expand to nearly the riverbank surface. The rainfall intensity greatly affected the depth of the wetting front and the height of the pond. In these cases, the rainfall intensity and changes in the saturation condition caused by the wetting front were the main factors affecting the FOS and the riverbank stability.

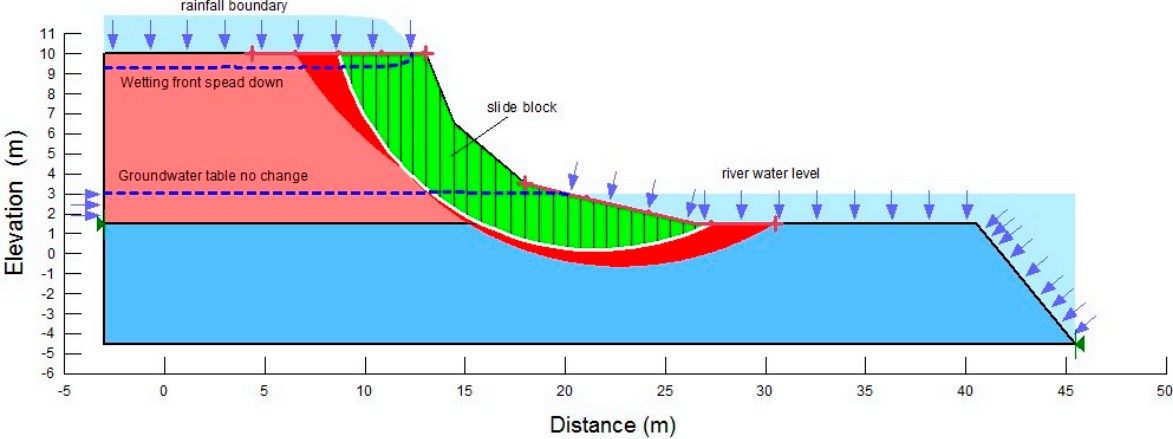

**Figure 12.** The wetting front descending when the RI is much higher than HC.

When the rainfall intensity was slightly higher than hydraulic conductivity (as in cases M-2-2 and M-2-3), the riverbank became saturated from both the groundwater table and the wetting front, as shown in Figure 13. Similar to the case of high hydraulic conductivity, the groundwater table rose by transient seepage through the unsaturated area. However, the groundwater rose at a slow rate. The wetting front appeared when excess rainwater was present on the surface. With only the rising of the groundwater table, the FOS decreased slowly, but the FOS changed more quickly when the wetting front descended. In this case, both the groundwater table and wetting front influenced the FOS and the riverbank stability.

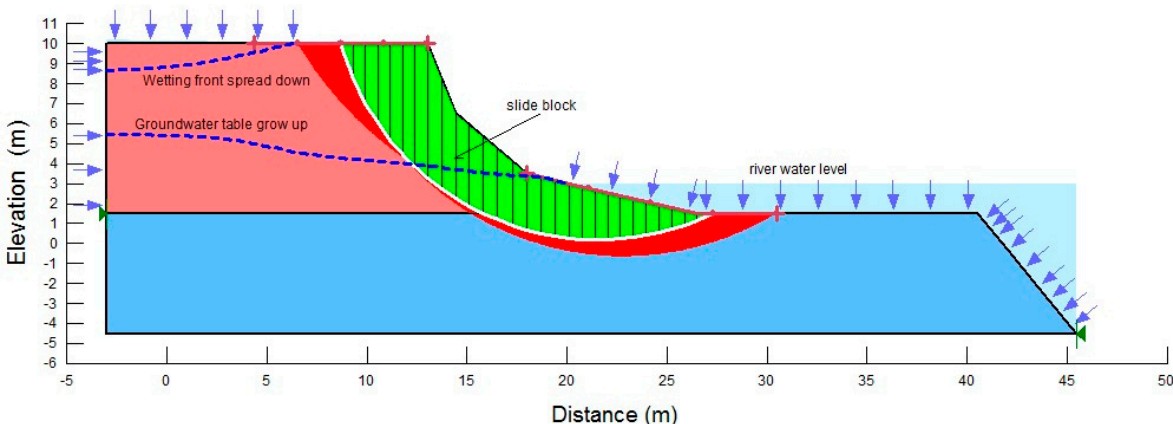

**Figure 13.** The rising of the groundwater table when the RI is lightly higher than HC.

In cases where the riverbank had good drainage conditions and no pond on the surface, the excess rainwater ran off the slope. When RI < HC (cases H-1-1, H-2-1, H-2-1, H-2-3, and M-2-1), the change of groundwater was the same as in the cases with a pond on the surface.

In cases with low hydraulic conductivity, (cases M-2-2, M-2-3, L-2-1, L-2-2, and L-2-3), the rainwater seepage occurred very slowly. The excess water ran off the slope, and the wetting front also spread slowly. In a short time, the groundwater table and the wetting front did not significantly increase. Then, the FOS slightly decreased (Figure 10C2). The decreasing of the FOS was due to the transient seepage change of the unsaturated soil properties as well as losing soil suction. This result also matched that in [2,5,19,22] when the rainfall intensity was smaller or equal to the soil hydraulic conductivity.

### 3.4. Effects of Rainfall Infiltration and River Water Level Fluctuation

The effects of multiple rainfall events and RWL in the rainy season are shown in Figure 14. The change in FOS was nearly the same as the change in RWL, which meant that the FOS increased with the increase in RWL, and the FOS decreased with the decrease in RWL. These results agreed with those of previous studies [39–42], which studied the effect of RWL on FOS. When the riverbank had a soil hydraulic conductivity lower than $10^{-4}$ cm/s, the FOS always depended on the change in RWL because of confining pressure [42]. The effect of rainfall on the riverbank was insignificant compared with the effects of changes in RWL. The results from the long-term analysis during the rainy season showed that low-intensity rainfall (less than 140 mm/day, average 5.8 mm/h) caused the FOS to decline at a low rate. Riverbank failure occurred only when the riverbank had a high slope and had cracks in the riverbank surface. In fact, there was a high slope angle and some cracks along the riverbank in the study area. In general, a riverbank with a high slope angle and some cracks has a high potential of failure during the rainy season. Moreover, the change in RWL was the factor with the greatest influence on the FOS of the riverbank. The riverbank was more damaged when multiple factors occurred together.

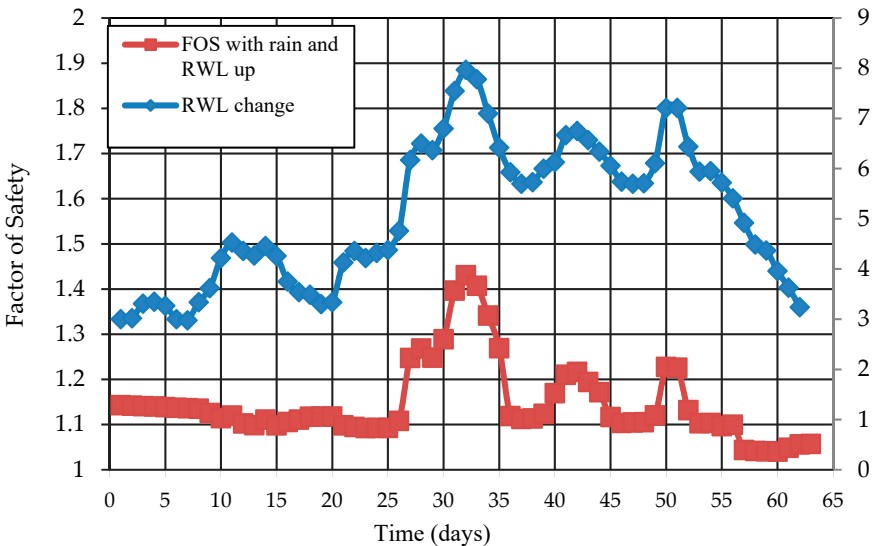

**Figure 14.** The FOS change with fluctuations of RWL and rainfall in the long-term.

## 4. Conclusions

The initial conditions such as the saturation degree and the groundwater table are the first important factors for riverbank slope stability in the rainy season. Setting the initial conditions leads to different mechanisms of infiltration, seepage, and changes of groundwater pressure.

In highly saturated conditions, the FOS decreases with the increase in rainfall intensity and accumulation. During a rainfall event, the rainwater infiltrates and affects the riverbank stability through two processes—changes in pore-water pressure and a wetting front controlled by the rainfall intensity and the hydraulic conductivity.

When the rainfall intensity is lower than soil hydraulic conductivity, the rainwater infiltrates, and transient seepage occurs through the unsaturated area, causing the groundwater table to rise. In soil with a higher hydraulic conductivity and rainfall intensity, the groundwater table rises quickly, leading to a higher potential of riverbank failure. Riverbank failure often occurs under high-intensity rainfall.

When the rainfall intensity is slightly higher than the soil hydraulic conductivity, the groundwater table rises slowly, and the FOS decreases slowly in the early period after raining. With a pond on the surface, the wetting front appears when there is excess rainfall water. The FOS decreases more quickly by the development of both the wetting front and the groundwater table convergence.

When the rainfall intensity is much higher than the soil hydraulic conductivity, the groundwater does not change in a short time. The wetting front descends slowly, and the height of the pond increases quickly with higher rainfall intensity. The wetting front and pond loading area are the main factors causing FOS changes.

In cases with no pond, the wetting front is on a shallow surface and descends very slowly, and the rainfall water transient seepage occurs slowly. In a short time, the groundwater table and the wetting front do not significantly increase; thus, the FOS slightly decreases. The decreasing of FOS is due to transient seepage changes of unsaturated soil properties as well as losing soil suction.

During long-term events with low-intensity rainfall, i.e., less than 10 mm/h, the FOS primarily depends on the river water level. A riverbank with a high slope angle and cracks with high hydraulic conductivity will have a higher potential of riverbank failure.

**Author Contributions:** Conceptualization, T.T.D. and D.M.D.; Methodology, T.T.D.; Software, T.T.D.; Validation, T.T.D., D.M.D. and K.Y.; Formal Analysis, T.T.D.; Investigation, T.T.D., D.M.D.; Resources, T.T.D.; Data Curation, T.T.D.; Writing-Original Draft Preparation, T.T.D.; Writing-Review & Editing, T.T.D.; Visualization, T.T.D.; Supervision, K.Y.; Project Administration, T.T.D.; Funding Acquisition, T.T.D.

**Funding:** This research was funded by the project Code 105.08-2015.24, which was sponsored by Nafosted, Ministry of Science and Technology, Vietnam.

**Acknowledgments:** This paper was completed with the support of the project Code 105.08-2015.24, which was sponsored by Nafosted, Ministry of Science and Technology, Vietnam, and the support by Geotechnical Laboratory, Ibaraki University, Japan to determine unsaturated soil properties. The authors express our sincere gratitude for these supports.

**Conflicts of Interest:** The authors declare no conflict of interest.

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
