# Peer review of "Assessing the Effects of Rainfall Intensity and Hydraulic Conductivity on Riverbank Stability"

_water, doi:10.3390/w11040741_

Round 1
Reviewer 1 Report
Review of “Assessing the Effects of Rainfall and Hydraulic Conductivity on Riverbank Stability by Couple Hydromechanical Simulation” submitted to Water for consideration for publication. The manuscript has significantly improved by the authors. I think with the exclusion of some figures and focus on the main objective, the manuscript has improved a lot. Still, the authors should significantly improve the English of the manuscript. I have made a few minor comments on that, but a very thoroughly review by someone with a full professional proficiency in English is needed. Below I have provided general and specific comments to the text. General comments In general, the language used is not sufficiently comprehensible and I would suggest that the authors get editing help from someone with a full professional proficiency in English. The abstract now start with a motivation statement, however, the abstract is now lacking a concluding remark. Please add a sentence to the abstract with the main conclusion of the manuscript. Specific comments Lines 9-11: The first sentence abruptly ends with “and transient”. Are the authors referring to “transient seepage”? Or something else? Please revise. Throughout the manuscript the authors use the term “transient”, but it is very unclear what is meant by that. Please carefully revise the whole manuscript regarding this term. Line 15: Please add “on” between “understanding” and “the mechanics”. Lines 16-18: Please replace “(1)” and “(2)” by “Firstly,“ and “Secondly,“. And add “And finally, when“ to line 20. Lines 16-18: Please change this sentence to “Firstly, when the rainfall intensity is lower than the soil hydraulic conductivity, the factor of safety (FOS) reduces by changes in the groundwater table, which is a result of rainwater infiltration and lateral flow through the unsaturated soil”. The last part I am just guessing, because I do not understand what the authors mean with “transient”. See also the first comment. Line 24: Please change “FOS only dues to” to “FOS is only due to”. Line 30: Please replace “interested” with “studied”. Line 57: RI and HC were already defined before. Please remove. Line 113: Please start a new sentence between “grain size” and “the saturate” Line 126: Please change “wide” to “widely used” Line 127: Please change “analyses slope and riverbank stability” to “slope and riverbank stability analysis” Line 140: What do the authors mean with “hydraulic dynamic”? Please clarify in the text. Line 194: Please change “15 kPa, 33 kPa” to “15 kPa and 33 kPa, respectively”. Lines 250-252: This was already mentioned before. Please remove.
Author Response
Responding Reviewers
Thank you very much for your recommendation for my paper.
I have revised following all your recommendations on contents and English writing. The English language is edited by MDPI. Please see detail in attach file
Thank you very much
Yours sincerely

Reviewer 2 Report
The paper is focused on the influence of rainfall characteristics on riverbank slope stability. In particular, the effect of soil hydraulic conductivity on the mechanics of rainfall infiltration is investigated with a commercial code by considering different initial saturation condition and surface drain condition. The study area is a riverbank site from the Red River in Hanoi, Vietnam. This topic is of interest for the readership of the Water journal.
This manuscript is a revised version of a previous paper: “Assessing the Effects of Rainfall and Hydraulic Conductivity on Riverbank Stability by Couple Hydromechanical Simulation” - Ref. N. 390131.
The presentation has been significantly improved following the recommendations of the three Reviewers. Anyway, English language should be carefully checked given that some typo and grammar errors can be found in the text.
In order to further improve the paper the Authors are recommended to consider the following requirements.
In the introductory discussion the Authors are recommended to add a sentence on the general aspects of slope stability and include the following references:
- Iverson, R.M. Landslide triggering by rain infiltration. Water Resour. Res. 2000, 36, 1897–1910.
- Lu, N.; Godt, J. Hillslope Hydrology and Stability; Cambridge University Press: New York, NY, USA, 2013.
- Manenti S., Amicarelli A., Todeschini S. (2018) WCSPH with limiting viscosity for modelling landslide hazard at the slopes of artificial reservoir. Water, 10(4), 515, doi: 10.3390/w10040515.
The “new” Figures 11, 12 and 13 are clearer than the previous version, anyway I suggest the Authors to provide a legend for these figures. Minor: Check the caption of Figure 13.
Author Response
Responding Reviewers
Thank you very much for your recommendation for my paper.
I have revised following all your recommendation on contents and English writing. The English language is edited by MDPI. Please see detail in attach file.
Thank you very much
Yours sincerely,
Duong Thi Toan

Round 2
Reviewer 1 Report
The manuscript has been significantly improved with respect to the English language. I have no further comments.
Reviewer 2 Report
The manuscript has been significantly improved following the reccomandations of the Reviewers; all my concerns have been addressed and convingcly justified.
This manuscript is a resubmission of an earlier submission. The following is a list of the peer review reports and author responses from that submission.
Round 1
Reviewer 1 Report
This paper is focused on the influence of rainfall characteristics on riverbank slope stability. In particular, the effect of hydraulic conductivity on the mechanics of rainfall infiltration is investigated with a commercial code. The study area is a riverbank site from the Red River in Hanoi, Vietnam
The manuscript is an original contribution and the topic is of interest for the readership of the Water journal.
English language should be carefully checked given that various typo and grammar errors can be found in the text. The Authors are recommended to respect the “consecutio temporum”.
The presentation is adequate; anyway, I have detected some criticisms in the text that should be properly addressed. “Materials and Methods” section should include also the description of the case study (and the adopted conditions). Results and Discussion should be merged into one section for a more effective presentation.
The Authors can benefit from the comments below to improve their paper. These have to be accomplished before manuscript acceptance.
Abstract
The abstract is concise and reflects the content of the article.
Line 14: Remove “causes”.
Introduction
Aims of the study are properly clarified in the Introduction.
Line 36: The Authors are recommended to complete the sentence as follows: “There are lots of interesting studies in technical literature dealing with the general aspects of slope stability (Lu et al. 2013; Iverson, 2000; Manenti et al. 2018) and the effects of rainfall properties on slope were interested by …”. Therefore the following references should be included in the introductory discussion:
- Iverson, R.M. Landslide triggering by rain infiltration. Water Resour. Res. 2000, 36, 1897–1910.
- Lu, N.; Godt, J. Hillslope Hydrology and Stability; Cambridge University Press: New York, NY, USA, 2013.
- Manenti S., Amicarelli A., Todeschini S. (2018) WCSPH with limiting viscosity for modelling landslide hazard at the slopes of artificial reservoir. Water, 10(4), 515, doi: 10.3390/w10040515.
Line 36: Replace “by the quite huge” with “by a quite huge”.
Line 50: Replace “specific” with “specifically”.
Line 58-61: “Simulated”? Perhaps “were also simulated”. Check the sentence.
Line 61-62: Check the time of verbs.
Line 64: Replace “the number of research” with “the number of researches” or “researches”.
Line 65: Replace “were still limited” with “are still limited”.
Line 65: Replace “there was” with “there are”.
Line 67: Replace “difference” with “differ”.
Line 69: Replace “hydraulic conduction will cause” with “hydraulic conductivity cause”.
Line 70: Replace “as mention” with “as mentioned”.
Line 71: Replace “the mechanics rainfall infiltration in difference soil” with “the mechanics of rainfall infiltration for different soil”.
Line 72: Replace “rain fall” with “rainfall”.
Line 75: Replace “the relationship the” with “the relationship of the”.
Line 76: Replace “in the difference saturation” with “in different saturation”.
Materials and Methods
Line 79: Replace “investigated data” with “experimental data”.
Line 83: Replace “investigates” with “investigations”.
Line 86: Replace “and included” with “including”.
Line 95: Replace “have had” with “had”.
Line 100: Replace “higher” with “high”.
Line 101-102: Replace “there has been damage” with “damage occurred”.
Line 102: Replace “Figure 2 shows the road was” with “Figure 2 shows that the road was”
Line 105: Replace “some problems current” with “some current problems”.
Line 108: Remove “it indicates that”.
Line 113: Check the first two rows of Table 1.
Line 118: Remove “were”.
Line 119 Replace “will be” with “is”. (Materials and Methods section)
120: Replace “with the change of the” with “for different”.
Line 123: Replace “predetermine” with “predetermined”.
Line 125: “were installed as difference”? Check the sentence.
Line 126: Replace “will be specific descripted” with “is specifically descripted”
Case study conditions
Line 133: I suggest including “Case study conditions” in the Materials and Methods” section.
Line 136-137: Remove “additionally, the riverbank is composed of homogeneous silty soil”. This sentence is repeated at lines 138-139.
Line 156: Remove “and will be used in this research”. Repetition.
Line 157: In Figure 7 daily RWL trend not always appears due to the daily rainfall. Please comment on this aspect.
Line 158: Replace “The change of” with “Daily”.
Line 164: Replace “;” with “,”.
Line 168: Replace “infiltrate” with “infiltrates”.
Line 180: Remove “Calculation” or add “of” after “Calculation”.
Line 188: Add “s” to “case Li”, i.e., “cases Li”
Line 193: Replace “difference unsaturation” with “different unsaturated”.
Line 201: In table 1, two columns indicate soil suction (kPa) with different values. Check the Table and correct.
Results
Results are presented in a logical sequence.
Line 218: Replace “three initial” with “three values of initial”.
Line 222: Replace “was apply in bothy case” with “was applied in both cases”.
Line 223-224: Check the text.
Line 256- line 260: In the captions of Figure 11, 13 and 14 replace “difference” with different.
I suggest indicating the riverbank failure in Figures 11, 13 and 14.
A legend should be provided for Figures 12,15 and 16
Line 261: Check the caption of Figure 15. I suggest “trend of the wetting front …” Caption should describe, not comment on the Figure.
Line 268: Replace “water infiltrate” with “water to infiltrate”
Line 276: Replace “FOS < 1” with “FOS became less than 1”
Line 279: Figure 16 has been cut. The x-axis label, i.e. Distance (m), is missing in the two graphs above.
Line 288: Replace “in the higher” with “for the higher”.
Line 289: Replace “in the different” with “for different”
Line 313: Replace “investigated” with “found”.
Line 323: Replace “when slope angle at” with “for slope angle of 80”.
Lines 326 and 329: Replace “in riverbank slope = 80” with “for riverbank slope of 80”.
Discussion
I suggest merging the section “Results” and the section “Discussion”. In my opinion a single section would be more effective and some repetitions would be avoided.
Line 332 Replace “with the difference of the initial” with “with different initial”.
Line 333: Replace “it is obviously show the” with “it is obviously shown that the”.
Line 334: Replace “high the saturation” with “high saturation”
Line 335-336: Reformulate the sentence.
Line 345: Add “s” i.e., “depends on”.
Line 379 Replace “and obviously and causes” with “and obviously causes”.
Conclusions
Conclusions seem reasonable and are supported by the results.
Line 395: typo error: replace “affects” with “effects”.
References
Three references are suggested in the Introduction section concerning important studies on the general aspects of slope stability, with a view to generality. Apart from these references, based on my knowledge, no important reference is missing.
Reviewer 2 Report
First, the manuscript needs an important improvement in the writing and wording of the text, as well as a general and extensive revision of the language (I recommended to contact with native English language). These two factors, in their current state, make the manuscript difficult to read and more difficult to understand, which significantly limits the possible interest of the exposed investigations. I think this is the key factor of the manuscript.
Most of the figures also need to be improved, since they lack aspects as important as legends, right vertical axes captions, grouping of several figures in only one figure, etc ...
The way in which bibliographic references are used does not consider it the most correct. I was surprised by the fact that in the introduction mention is made of the bibliographical references in forms such as: [2-29] or [2-20]. And yet, there is no bibliographic reference cited in the Discussion section. This last point can not be accepted, so you must to compare your results with previous ones or the general accepted ideas about your topic.
Table 2 presents two columns with the same title and that nevertheless collect different values, which is surprising.
Abbreviations are used that have not been explained before.
It is essential to improve the exposure of the Results section, as they are confusing. In the Discussion, new data appear that have not been mentioned previously. The Conclusion is repetitive with respect to aspects that have already been mentioned, without clarifying or grouping the main results offered by the research carried out.
Reviewer 3 Report
Review of “Assessing the Effects of Rainfall and Hydraulic Conductivity on Riverbank Stability by Couple Hydromechanical Simulation” submitted to Water for consideration for publication.
The manuscript describes a sensitivity analysis of a river bank failure model. I think it is an important topic to study and it seems that the study is well-performed. The authors studied a variety of model variables, such as soil hydraulic conductivity, rainfall intensity, pore pressure, slope angle, ponding and cracks. Because of the large amount of variables, the manuscript is lacking a clear focus. From the beginning of the manuscript, it appears that the focus is on soil hydraulic conductivity and rainfall intensity. I suggest to focus the paper on these two variables. See below for some other comments regarding this issue.
The objective of the study is to understand the impact of rainfall intensities and differences in soil hydraulic conductivity to bank slope stability. The authors have studied this by performing several scenarios with changes in rainfall intensity and soil hydraulic conductivity. However, also several other things have been studied, such as the effect of changes in the bank slope, with and without cracks and ponding, and negative pore pressure. Because the authors vary so many variables, the manuscript is very difficult to follow. I suggest to keep the focus on rainfall intensity and soil hydraulic conductivity and remove the other variables from the analysis, especially the variation in bank slopes and the scenarios with and without cracks. The other variables (i.e. negative pore pressure and ponding) are somehow related to the rainfall intensity and soil hydraulic conductivity and may be useful to include in the analysis. This would considerably increase the readability of the manuscript.
Apart from the soil properties and bank geometry, the authors did not use field data to calibrate or validate the model. The rainfall intensities were obtained from analysis of rainfall timeseries and a hydraulic conductivity was obtained from field measurements. But it is not clear why the authors choose the values with respect to the variation in degrees of saturation and negative pore pressure. These are shown in lines 177-178, but the values are not substantiated. The same holds for the slope angles. Please include a small discussion why these values are used, because at the moment they seem arbitrary. But this can be neglected when the authors choose to follow my suggestion to focus solely on soil hydraulic conductivity and rainfall intensity.
Overall, I think this manuscript has potential to be a valuable addition to the literature, but some works is still required to make it acceptable for publication.Below I have provided general and specific comments to the text.
General comments
1. In general, the language used is not sufficiently comprehensible and I would suggest that the authors get editing help from someone with full professional proficiency in English.
2. The abstract normally starts with 1 or 2 sentences of motivation. Please include a motivational statement to the first part of the abstract.
3. In the Introduction, the authors refer to infiltration and how saturation and rainfall intensity play a role in this respect. It would be good to extent the Introduction slightly with a discussion of the different infiltration mechanisms, i.e. saturation and infiltration excess. See for instance Beven (2012) for a description of the differences between the two main infiltration mechanisms.
4. The manuscript has a lot of figures (22!). Please consider to include some of the less important figures in the Supplementary Material, such as figures 4, 5, 6. Besides some of the figures may be summarized into one figure. I suggest to combine figures 11, 13 and 14 in one figure (i.e. 6 panels) with similar y-axis limits to facilitate comparison between the scenarios (now in figures 13 and 14 the y-axis of the left and right panels are not the same and, therefore, it is difficult to see the difference in FOS response).
5. The Discussion section needs to be significantly revised. As far as I understand it well, the results from figures 11, 13 and 14 are shown in two new figures (21 and 22). Please remove these redundant figures and refer to figures 11, 13 and 14 when discussing the results. Furthermore, the results should be discussed in line with previous research. For instance, how does this study compare to previous studies with the same model (Geoslope), what do previous studies say on the sensitivity of soil hydraulic conductivity and rainfall intensity, etc.
Specific comments
Line 10: The authors mean “This paper has the objective”?
Line 14: The authors mean “causes an increase in”?
Lines 26-35: The authors refer to previous research in this paragraph, but there is only reference made to 1 article. Please include references to relevant literature.
Lines 51-52: The authors mean “10-4” and “10-6”?
Lines 61-62: Please replace this sentence with a conclusion from [27] and [28].
Lines 87-88: The groundwater level and river water level were not collected at the field sites? Please clarify in the text.
Line 118: Replace “;” with “,”.
Line 118: What do the authors mean with “hydraulic dynamic”? Please clarify in the text.
Line 122: SEEP/W is a module of the Geo-slope model (I guess). Please provide a short description of this module.
Line 131: Please define FOS in the previous sentence, i.e. “factor of safety (FOS)”.
Lines 137-138: Why is the riverbank also analyzed with higher angles? Please clarify in the text.
Lines 140-142: “the unsaturated…Geoslope program.” This was already shown in lines 93-94, please remove this sentence.
Lines 217-224: This paragraph is redundant and merely summarizes the Material & Methods. Please be concise and don’t be repetitive. I suggest to remove this paragraph.
Line 225: As I understand it well, the combination S = 87% and P = -15kPa is the reference for degree of saturation and pore pressure. Please indicate this in the Material & Methods.
Lines 263-265: Which figure or table shows these results? Please clarify in the text.
Lines 305-316: This is a discussion of the results, please move to the Discussion.
Figures and Tables
Figure 1: Please describe in the caption what the is shown in the two panels, e.g. what do the blue dots and the red line in the left panel represent and what do the black lines in the right panel represent?
Figure 7: Rainfall is most often represented with a bar chart, rather than a line chart. Please consider this for this figure. Please remove the markers (red squares and blue diamonds).
Figures 11, 13, 14: Please indicate the scenario (with and without ponding) in each of the two panels.
Figures 12, 15, 16: Please discuss in the caption what the different lines and colors represent. The blue dashed line is the groundwater table? Please indicate this in the caption. But also the blue arrows, the green shaded area, the red area, the yellow area and the green area.
Figure 16: Please indicate in each panel to which number of days each panel corresponds.
Figure 21 and 22: Do these figures show the same results as figures 11, 13 and 14? If so, please remove these figures and refer to figures 11, 13 and 14.
References
Beven, K. J.: Rainfall-runoff modelling: the primer, John Wiley & Sons, Ltd, 2012.